# Circulating PCSK9 Linked to Dyslipidemia in Lebanese Schoolchildren

**DOI:** 10.3390/metabo12060504

**Published:** 2022-05-31

**Authors:** Yara Azar, Marie-Hélène Gannagé-Yared, Elie Naous, Carine Ayoub, Yara Abou Khalil, Elise Chahine, Sandy Elbitar, Youmna Ghaleb, Catherine Boileau, Mathilde Varret, Petra El Khoury, Marianne Abifadel

**Affiliations:** 1Laboratory of Biochemistry and Molecular Therapeutics (LBTM), Faculty of Pharmacy, Pôle Technologie-Santé (PTS), Saint-Joseph University of Beirut, Beirut 175208, Lebanon; yara.azar@inserm.fr (Y.A.); carine.ayoub@net.usj.edu.lb (C.A.); aboukhalilyara@gmail.com (Y.A.K.); sandyelbitar@gmail.com (S.E.); youmna.ghaleb@inserm.fr (Y.G.); petra.el-khoury@inserm.fr (P.E.K.); marianne.abifadel@usj.edu.lb (M.A.); 2Laboratory for Vascular Translationnal Science (LVTS), INSERM U1148, Bichat Hospital, F-75018 Paris, France; catherine.boileau@inserm.fr (C.B.); mathilde.varret@inserm.fr (M.V.); 3Centre Hospitalo-Universitaire Xavier Bichat, Paris Cité University, F-75018 Paris, France; 4Department of Endocrinology, Faculty of Medicine, Saint-Joseph University of Beirut, Beirut 175208, Lebanon; 5Division of Endocrinology, Hôtel-Dieu de France Hospital, Beirut 166830, Lebanon; elie.naous@net.usj.edu.lb (E.N.); elise.adel.chahine@gmail.com (E.C.); 6Genetic Department, AP-HP, Centre Hospitalo-Universitaire Xavier Bichat, F-75018 Paris, France

**Keywords:** pediatric, dyslipidemia, PCSK9, lipoprotein (a), cholesterol, Lebanon

## Abstract

In adults, elevated levels of circulating Proprotein Convertase Subtilisin/Kexin type 9 (PCSK9) have been associated with increased Low-density lipoprotein cholesterol (LDL-C), triglycerides (TG), and worse cardiovascular outcomes. However, few studies analyzed the relation between PCSK9 and lipid parameters in pediatric populations. The aim of our study is to evaluate the distribution and the correlation of serum PCSK9 levels with lipid parameters in a sample of Lebanese school children. Using an immunofluorescence assay, we measured serum PCSK9 levels in 681 school children recruited from ten public and private Lebanese schools. We analyzed the association between PCSK9 and age, sex, Body Mass Index (BMI), and lipid parameters (total cholesterol (TC), LDL-C, TG, High-density lipoprotein cholesterol (HDL-C), non-HDL-C, and lipoprotein (a) (Lp(a)). Serum PCSK9 levels were significantly correlated with TC, LDL-C, and non-HDL-C (*p* value < 0.0001) but not with TG, HDL-C, and Lp(a). PCSK9 levels were also significantly higher in children with high TC, LDL-C, and non-HDL-C (*p* values = 0.0012, 0.0002, 0.001, respectively). No significant gender differences in PCSK9 were found. In addition, no significant associations between PCSK9 and both age and BMI percentiles were observed. In girls, no difference in PCSK9 values was observed according to menarche while in boys, testosterone levels were not significantly associated with PCSK9. Serum PCSK9 levels were significantly correlated with TC, LDL-C, and non-HDL-C levels. Further studies are needed to find if PCSK9 measurements have an additional value to predict future cardiovascular outcomes in pediatric populations.

## 1. Introduction

Atherosclerotic cardiovascular disease (ASCVD) is a leading cause of mortality in the world [1]. Atherosclerotic lesions begin early in life and are highly correlated with dyslipidemia [2,3,4]. At an early age, high low-density lipoprotein cholesterol (LDL-C) accelerates the onset of atherosclerosis leading to coronary heart disease (CHD) in adulthood [5]. Early diagnosis and optimal management of abnormal lipid profile will likely slow down the progression of atherosclerosis [6]. Lipoprotein(a) (Lp(a)) is another lipoprotein associated to ASCVD [7]. It has a similar composition to LDL-C and is characterized by the presence of a covalent bond between apolipoprotein(a) (apo (a)) and apolipoprotein(B) (apo B100) [8].

The Proprotein Convertase Subtilisin/Kexin type 9 (PCSK9) gene was identified in 2003 by Abifadel et al. as the third gene responsible for familial hypercholesterolemia (FH) [9]. The encoded protein of 692 amino acids is an enzyme belonging to the large family of proprotein convertases, and it is mainly expressed in the liver, gut, kidney, and nervous system [10]. PCSK9 degrades the LDL receptor (LDLr) by binding to its epidermal growth factor-like domain A (EGF-A) independently of its catalytic activity and directs the LDLr into lysosomes for degradation [10]. The PCSK9 gain of function variants are associated with elevated levels of LDL-C and are responsible for some forms of FH and an increase in coronary heart disease [11], while loss-of-function variants are associated with low LDL-C levels and a decrease in ASCVD risk [12]. In adult populations, several studies have evaluated the association between PCSK9 levels and other lipid or inflammatory parameters [13,14,15,16]. Elevated levels of circulating PCSK9 have also been associated with increased LDL-C, triglycerides (TG), and worse cardiovascular outcomes [17]. However, only one study has been conducted on children based on a French-Canadian population sample [18].

Dyslipidemia refers to a group of lipid disorders characterized by high levels of total cholesterol (TC), LDL-C, non-high-density lipoprotein cholesterol (non-HDL-C), TG, and low levels of high-density lipoprotein cholesterol (HDL-C) [19]. Non-HDL-C, which is calculated by subtracting HDL-C from TC, is as good as or better than other lipid measures, such as LDL-C, in predicting atherosclerosis. In addition, because in children the difference between fasting and nonfasting TG levels is minimal, a nonfasting measurement of TG could be reliable [20].

Dyslipidemia is highly prevalent in Lebanon in both adults [21,22] and children [19,23,24]. In our previous study, we found that the prevalence of high non-HDL-C, high TG, and high Lp(a) in Lebanese school children are, respectively, 9.2%, 26.6%, and 14.4% [23,24]. However, PCSK9 has never been measured in Middle Eastern pediatric populations. 

The objective of this study is to evaluate circulating PCSK9 levels in a sample of the Lebanese pediatric population and to characterize its association with age, gender, BMI, lipid parameters, and Lp(a). In addition, the relationship of PCSK9 with menarchal status in girls and with testosterone in boys is assessed.

## 2. Results

### 2.1. Characteristics of the Population

A total of 681 children, 347 boys and 334 girls (50.95% boys and 49.05% girls) were included in the study (Table 1). The median age of the population was 12.94 (10.68–14.73) years with no significant gender difference (*p*-value = 0.19). As shown in Table 1, 40.97% of subjects are aged between 8–11 years, 35.7% between 12 to 15 years, and 23.3% between 15 and 18 years. A total of 183 girls (54.80%) had their menarche.

The median BMI percentile was 69.50 (38.00–88.90) with no significant difference between boys and girls [70.20 (38.20–90.70) vs. 68.60 (37.70–85.35), respectively, *p*-value = 0.23]. 11.31% (13.26% boys and 9.28% girls) of the participants were obese and 19.09% (21.90% boys and 16.17% girls) were overweight, with a significant gender difference when comparing the 3 BMI groups (*p*-value = 0.02).

### 2.2. Distribution of Lipid Parameters According to Age and Gender

Lipid parameters in the entire population and by gender are also shown in Table 1. Girls had significantly higher TC, non-HDL-C, and TG levels than boys (*p* values of 0.02, 0.048, and 0.03, respectively), while no significant gender differences are observed for HDL-C and LDL-C. No significant difference was observed in Lp(a) between girls and boys (*p*-value = 0.34). 19 children (2.8%) had very high LDL-C (≥4.1 mmol/L) suggestive of FH [25]. However, in these children, we did not collect information on their family history of premature coronary artery disease or perform a DNA analysis to confirm the diagnosis of FH.

### 2.3. PCSK9 Levels Distribution in the Overall Population

In the overall population, serum PCSK9 levels varied from 13.48 to 692.9 ng/mL (Figure 1) with no significant gender difference (*p*-value = 0.60) as shown in Table 2. No significant correlation was observed between PCSK9 and both age and BMI percentiles in the entire sample (r = −0.013, *p* = 0.74; r = 0.035, *p* = 0.36, respectively), and when the analysis was performed separately in boys and girls (For age: [r = −0.079, *p* = 0.14 in boys and r = 0.057, *p* = 0.30 in girls]). For BMI percentiles: [r = 0.08, *p* = 0.14 in boys and r = −0.036, *p* = 0.51 in girls, respectively]) (Table 2).

### 2.4. PCSK9 Relationship with Lipid Parameters

In the total population, PCSK9 levels correlated significantly, with TC, LDL-C and non-HDL-C levels: r = 0.198, *p* < 0.0001; r = 0.196, *p* < 0.0001; and r = 0.185, *p* < 0.0001, respectively. This correlation was still significant when boys and girls were analyzed separately: r = 0.19, *p* = 0.0003; r = 0.19, *p* = 0.0003; and r = 0.18, *p* = 0.0011, respectively, for boys, and r = 0.21, *p* < 0.0001; r = 0.21, *p* < 0.0001; and r = 0.21, *p* < 0.0001, respectively, for girls.

However, in the overall population as well as in boys and girls, no significant correlation between PCSK9 and TG was found (r = 0.037, *p* = 0.33; r = 0.038, *p* = 0.49; and r = 0.047, *p* = 0.39, respectively). Similarly, no significant correlation was found between PCSK9 and HDL-C (r = 0,028, *p* = 0.46; r = 0.055, *p* = 0.31; and r = 0.00038, *p* = 0.99, respectively), or Lp(a) (r = −0.022, *p* = 0.56; r = −0.067, *p* = 0.21; and r = 0.034, *p* = 0.53, respectively).

### 2.5. Comparison of PCSK9 Levels between Normolipidemic and Hyperlipidemic Subgroups

In the overall population, 53 subjects (7.78%) had high TC, 87 (12.78%) had high LDL-C, 72 (10.57%) had high non-HDL-C, and 198 (29.07%) had high TG. The PCSK9 levels were 31.80%, 26%, and 18.7% higher in the subgroups with high TC, high LDL-C, and high non-HDL-C, respectively, compared to the normal/borderline subgroups (*p* = 0.001, *p* = 0.0002, *p* = 0.001, respectively) (Figure 2). No significant difference was found in PCSK9 levels between the subgroup with high TG compared to the one with normal/borderline TG (*p*-value = 0.86) (Table 3).

Finally, PCSK9 levels did not differ between the 88 children (12.92%) with abnormal Lp(a) levels and the rest of the sample (*p*-value = 0.23) (Table 3).

### 2.6. Relationship between PCSK9 and Menarche in Girls and Testosterone in Boys

The PCSK9 levels did not differ significantly between girls who already had their menarche (*n* = 183) and those who did not (*n* = 151) (61.85 ng/mL (46.64–89.65) vs. 62.32 ng/mL (46.36–87.81), *p*-value = 0.94).

A significant positive correlation between serum PCSK9 levels and TC, LDL-C and non-HDL-C was observed in post menarche (r = 0.323, *p* < 0.0001; r = 0.30, *p* < 0.0001; r = 0.31, *p* < 0.0001, respectively), but not in pre-menarche (r = 0.077, *p* = 0.34; r = 0.071, *p* = 0.38; r = 0.071, *p* = 0.39, respectively).

In boys, the correlation between serum PCSK9 levels and testosterone levels for values above the detectable threshold of >20 ng/mL (*n* = 216) was not significant (r = −0.014, *p*-value =0.84).

### 2.7. Multiple Linear Regression Analysis with PCSK9 as the Dependent Variable

Using multiple linear regression, PCSK9 remained significantly associated with non-HDL-C in both boys and girls (respective *p*-values = 0.0003 and 0.0001), while no independent association was found between PCSK9 and age, BMI, TG, HDL-C (Table 4).

## 3. Discussion

Our results showed a positive correlation between PCSK9 and TC, LDL-C, and non-HDL-C levels in a sample of 681 Lebanese school children. This finding was observed in both boys and girls. The same result was also found in a sample of 1736 French-Canadian children and adolescents aged 9, 13, and 16 years [18]. Other studies have also demonstrated that serum PCSK9 levels were significantly correlated with TC concentrations and serum LDL-C in adults [16,17]. Proprotein Convertase Subtilisin/Kexin type 9 is one of the ligands of LDLr; it uses it as an exit route from the plasma compartment while also inducing its degradation; thus, plasma PCSK9 levels, LDLr expression, and plasma LDL-C levels are all reciprocally regulated [26]. The PCSK9 half-life was shown to be much longer in LDLr ^−/−^ mice, while overexpression of hepatic LDLr decreases serum levels of murine PCSK9 [27]. The LDL particle also increases PCSK9 affinity for the LDLr by protecting its cleavage from furin; this results in a substantial decrease in plasma PCSK9 activity [26].

Interestingly, we found significantly higher PCSK9 levels among children with high TC, LDL-C, and non-HDL-C groups compared to those with lower levels. As far as we know, our study is the first to report this association. It has already been shown in adults that PCSK9 levels were higher in untreated homozygous or heterozygous FH patients compared to controls and that there was a correlation between PCSK9 levels and LDL-C in untreated subjects with FH [15,28]. It is noteworthy that another study has demonstrated that PCSK9 levels could be related to cardiovascular (CV) events independently of the levels of atherogenic lipoproteins [29]. Finally, Leander et al. showed that even after adjusting for known CV risk factors, the association between serum PCSK9 and CV risk remained, suggesting that PCSK9 levels may play a role in CV events apart from regulating LDL cholesterol [30].

We did not find any correlation between both TG, HDL-C levels, and serum PCSK9 concentrations, although Baass et al. reported in a French-Canadian pediatric cohort a correlation between PCSK9 and both TG and HDL-C levels in boys and girls [18]. Lakoski et al. also found in adults a positive association between PCSK9 and TG in a large ethnically diverse population [17]. On the opposite, and similarly to our report, Lambert et al. did not show this correlation [31]. Ethnic differences, non-fasting samples, or the smaller sample of our population could explain these differences.

We also did not find any correlation between PCSK9 and Lp(a). The correlation between PCSK9 and Lp(a) has never been studied in children. In adult populations, PCSK9 has been found to be positively correlated with Lp(a) [32]. The absence of a relationship between Lp(a) and PCSK9 can be explained by the highly genetic determination of Lp(a). It is also possible that the difference observed between adults and children is related to sex steroids or ethnicity.

Furthermore, we did not find any relationship between PCSK9 and age, gender, or BMI. A gender and age difference in PCSK9 levels was reported in the French-Canadian study [18]. The latter found that PCSK9 values were significantly higher in 9-year-old boys than in 13 and 16-year-old boys, whereas the inverse was true in girls. They also reported no significant relationship between PCSK9 levels and BMI in both boys and girls, which is similar to our findings [18]. On the opposite, higher BMI was associated with higher PCSK9 levels in an obese Sub-Saharan African adult population [33]. While in the present report no significant relationship was noted between PCSK9 and BMI, it is noteworthy that in our previously published results on the same cohort, BMI was significantly correlated with non-HDL-C and TG [23]. This might suggest that PCSK9 is more genetically determined compared to the other lipoproteins.

Finally, we looked at the relation between PCSK9 and pubertal status. We did not find any significant difference in PCSK9 levels according to menarchal status. Estradiol (E2) has been shown to regulate circulating PCSK9. In fact, in vitro studies found that E2 increases *LDLr* gene expression by 4 to 5-fold and downregulates the hepatic and plasma PCSK9 levels through the decrease in sterol regulatory element-binding protein-2 (SREBP-2) mRNA levels, resulting in a decrease in both PCSK9 and LDL-C levels [34,35]. In premenopausal women, PCSK9 levels were found to be inversely associated with E2 levels, with the lowest values occurring during ovulation, when E2 levels are at their highest levels [36]. The absence of difference that we found in PCSK9 between pre- and post-menarche girls may be explained by the fact that E2 levels at the starting stages of puberty are not sufficiently high enough to induce a decrease in PCSK9 levels. In addition, E2 levels start to increase well before the onset of menarche leading to a wide variation of E2 levels around the timing of menarche. We also interestingly found in post-menarchal girls, a significant positive correlation between serum PCSK9 levels and TC, LDL-C, and non-HDL-C a finding not observed in premenarchal girls. This finding can be explained by the change in lipid profile with age as we previously reported in the same cohort [24]. In this study, we found that 53.6% of school children have normalized their abnormal lipid profile after a 3-years follow-up. Therefore, stable levels of PCSK9 across time as opposed to the variability in the lipid profile could explain the different relationship between PCSK9 and lipid profile in pre- and post-menarchal girls. Similarly, in boys, we found no correlation between testosterone and serum PCSK9 levels. In another previous report, testosterone levels were also not associated with PCSK9 or LDL-C in healthy adult individuals [37].

The positive association we found between PCSK9 and TC, LDL-C, and non-HDL-C are important for the potential use of PCSK9 as a predictor of cardiovascular (CV) risk in children. Previous studies have found that elevated circulating PCSK9 levels in adults are predictors of CV events and are independently associated with arterial stiffness [38]. It was also speculated that PCSK9 can be used as a possible early marker of CV risk in healthy adults [29]. Extending these studies to the pediatric population would be also interesting since pediatric dyslipidemia leads to early atherosclerosis in autopsy studies [6]. Moreover, approximately one-half of children with abnormal lipoprotein values continue to have elevated lipid levels in adulthood [6,39,40]. The PCSK9 gene has been associated with inflammation in several studies as reviewed by Momtazi-Borojeni et al. Furthermore, the subendothelial accumulation of cholesterol caused by hypercholesterolemia leads to vascular inflammation [41]. This highlights the importance of identification and management of risk factors to slow down the progression towards atherosclerosis that begins early in childhood and in young adult life [6]. Further prospective studies are needed on a larger scale in different populations to confirm if PCSK9 levels provide additional information in predicting CVD beyond the recognized CV factors.

The current study has several limitations: first, the children’s sampling was not performed in a fasting state. However, circulating PCSK9 seems to be poorly affected by a fasting state since in healthy adults, plasma PCSK9 concentrations vary minimally in response to a short-term high-fat diet [42]. Second, estradiol was not measured in girls to examine the relationship between PCSK9 and estradiol. Third, no data were collected on the food intake of the recruited children since it has been demonstrated that PCSK9 can be affected by diet [43]. Furthermore, the cross-sectional nature of our study precludes any causality link between PCSK9 and the other components of the lipid profile. Finally, the lack of a standardized PCSK9 assay could explain the discrepancies in plasma PCSK9 levels reported in different studies.

## 4. Materials and Methods

### 4.1. Population

The population included in this study is a subgroup of our initial cohort. In our previous cross-sectional reports, 969 school children aged 8 to 18 years were recruited from 10 different private and public Lebanese schools using a stratified random sampling [23,44]. The schools were selected in the Greater Beirut and Mount Lebanon regions, where most of the Lebanese population is concentrated. Recruitment was conducted between May 2013 and October 2014.

Exclusion criteria were: any acute or chronic medical condition (such as diabetes or hypothyroidism) or intake of drugs that can affect the lipid profile (contraceptive pills, isotretinoin, oral corticosteroids, atypical antipsychotics, or immunosuppressive therapies).

The same device was used to measure height and weight on the day of sampling. Body Mass Index (BMI) was calculated by dividing weight in kilograms by height in meters squared (kg/m^2^). To account for age and sex differences, all BMI measurements were compared to age and sex-specific reference values from the Centers for Disease Control and Prevention (CDC) growth charts from 2000 to determine weight status [45]. This comparison was chosen because of the scarcity of reference values in Lebanon. BMI for sex and age below the 85th percentile was classified as normal, overweight was defined as BMI for sex and age between the 85th and 95th percentile, and obesity was defined as BMI for sex and age over the 95th percentile [45]. The population was categorized into three age groups: 8–11 years, 12–14 years, and 15–18 years as performed in our previous publications [23,44].

Blood samples were collected in 2013 and 2014 from non-fasting children between 8 and 10 AM. Serum was then stored at −80 °C until PCSK9 measurements in 2020. From the initial cohort, 288 children were excluded due to insufficient serum volume to perform PCSK9 measurement [23].

Based on the lipid thresholds established by the NHLBI [46], lipid values were defined as follows: (1) for TC: normal (<4.4 mmol/L), borderline (4.4–5.2 mmol/L), high (≥5.2 mmol/L), (2) for LDL-C: normal (<2.8 mmol/L), borderline (2.8–3.4 mmol/L), high (≥3.4 mmol/L), (3) for non-HDL-C: normal (<3.1 mmol/L), borderline (3.1–3.8 mmol/L), high (≥3.8 mmol/L), (4) for TG: normal (<0.8 mmol/L for children ≤9 years and <1 mmol/L for children >10 years, borderline (0.8–1.1 mmol/L for children ≤9 years, 1–1.5 mmol/L for children >10 years) high (>1.1 mmol/L for children ≤9 years and >1.5 for children >10 years). Based on epidemiological and Mendelian randomization studies, we defined the fifth group characterized by high Lp(a) (≥75 nmol/L) [47,48,49]. 

Informed consent was signed by all the parents’ children. Subjects aged 18 years signed the consent form as they do not need the consent of their parents. Children who refused blood sampling were excluded from the study. The study was approved by the ethics committee of Hôtel-Dieu de France university hospital, Beirut, Lebanon (2013 CEHDF 518, 2018 CEHDF 1174).

### 4.2. Biological Parameters

#### 4.2.1. Serum PCSK9 Assay

The PCSK9 measurements were performed using a commercial ELISA Kit (Human Proprotein Convertase 9/PCSK9 Duoset catalog no. DY3888, R&D Sytems, Minneapolis, MN, USA) and the Luminex Bio-Plex Pro assay technology as previously described [50]. This assay is based on the use of carboxy-coated Bio-Plex ProTM COOH beads (catalog no. MC10035-01, Bio-Rad, Hercules, CA, USA) covalently coupled with a PCSK9 capture antibody using the Bio-Plex Kit Amine Coupling Kit (catalog no. 171-406001, Bio-Rad, Hercules, CA, USA) according to manufacturer’s instructions. Next, the diluted samples were incubated with the coupled beads before the PCSK9 detection antibody was added. Then, streptavidin-phycoerythrin conjugate (Bio-Plex Streptavidin-PE catalog no.171304501, Bio-Rad, Hercules, CA, USA) was added to form the final detection complex. The intensity of the fluorescence signal was acquired using a Luminex-based reader (Bio-Plex 200 Systems, Bio-Rad, Hercules, CA, USA) [50].

#### 4.2.2. Lipid Profile

The measurements of TC, HDL-C, and TG were performed on a Vitros 5.1 FS machine (Ortho-Clinical Diagnostics, Inc., Raritan, NJ, USA) as previously described [23]. Non-HDL-C was calculated by subtracting HDL-C from the TC. LDL-C was calculated using the Friedewald formula where LDL-C, TC, and HDL-C are expressed in mmol/L (LDL-C=TC-(HDL-C + TG/2.2)).

#### 4.2.3. Lipoprotein (a)

The Lp(a) measurement was performed using an immunoturbidimetric assay (Cobas Integra 400 plus system, Roche Diagnostics, Basel, Switzerland) as previously described [44]. In this procedure, human Lp(a) agglutinates with latex particles coated with anti-Lp(a) antibodies. Turbidimetry at 659 nm is used to determine the precipitate. According to the manufacturer, the threshold value which indicates an increased risk is 75 nmol/L. The intra and inter-coefficient of variation is less than 4%.

#### 4.2.4. Total Testosterone

Total testosterone levels were measured only in boys using the Immulite 2000 automate (Siemens, Berlin, Germany). The sensitivity of the method is 20 ng/mL and the coefficient of variation was < 10% for values > 150 ng/mL.

#### 4.2.5. Statistical Analysis

The variables were analyzed using the GraphPad Prism version 9. Quantitative variables were not normally distributed using the Kolmogorov–Smirnov statistic to test for normality. Results for quantitative variables were expressed as median with its interquartile ranges (1st quartile and 3rd quartile). The Chi-square test and Mann–Whitney U test were used to perform univariate analysis. Spearman correlation was performed to measure the strength and direction of a linear relationship between PCSK9 and other quantitative variables. A Kruskal–Wallis test was used to compare the lipid parameters according to age and BMI groups. Two separate multiple linear regression analyses were performed in boys and girls with natural logarithm (Ln (PCSK9) as a continuous dependent variable and age, Ln(BMI), Ln(non-HDL-C), Ln(TG), Ln(Lp(a)) as independent variables. In the regression, TC, and LDL-C were not entered as independent variables because TC, LDL-C, and non-HDL-C are very strongly interdependent. Non-HDL-C was chosen as an independent variable because the sampling of children was performed in a non-fasting state and non-HDL-C has been recommended as a better measurement, especially among subjects with high TG levels. In addition, non-HDL-C is considered as good as or better than other atherogenic lipoproteins in predicting adult dyslipidemia and subclinical atherosclerosis [51].

## 5. Conclusions

We found a positive correlation between serum PCSK9 levels and TC, LDL-C, and non-HDL-C levels and higher PCSK9 levels among children with high TC, LDL-C, and non-HDL-C, while no relationship was observed between PCSK9 and TG, HDL-C or Lp(a). Age, gender, and BMI were also not found to be associated with PCSK9. Since high levels of PCSK9 play a significant role in the progression of atherosclerotic lesions, measuring PCSK9 in children, particularly those with high CV risk, could be useful to predict the risk of future CV events. To confirm this, further studies are needed to fully understand the genetic and metabolic factors that influence circulating PCSK9 levels in children.

## Figures and Tables

**Figure 1 metabolites-12-00504-f001:**
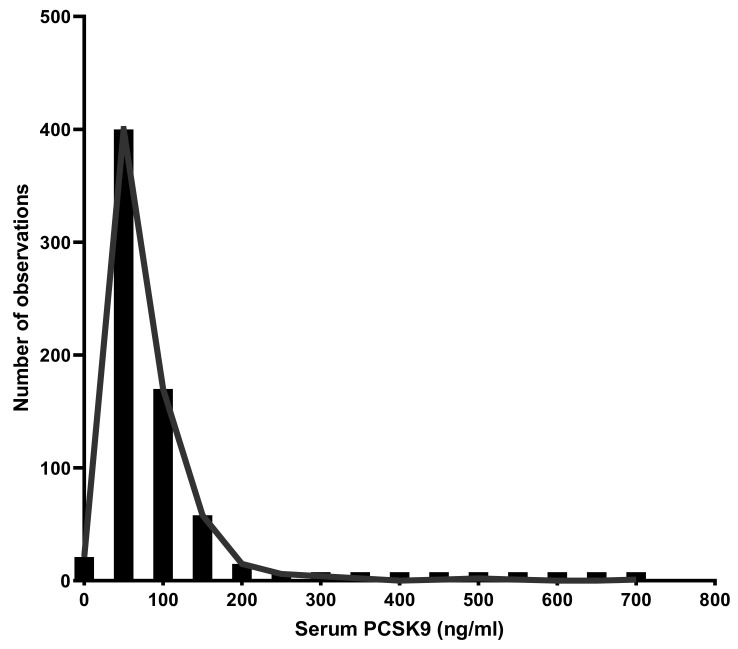
The distribution of serum proprotein-convertase-subtilisin/kexin type 9 (PCSK9) in the overall population.

**Figure 2 metabolites-12-00504-f002:**
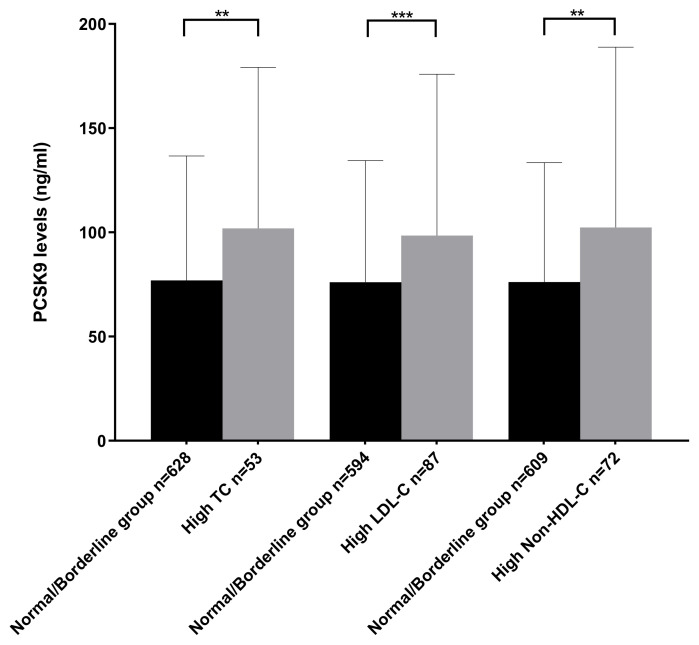
Median PCSK9 levels in the normal group and in the three subgroups [high TC (≥5.2 mmol/L), high LDL-C (≥3.4 mmol/L), high Non-HDL-C (≥3.8 mmol/L)]. ** *p* = 0.001, *** *p* = 0.0002.

**Table 1 metabolites-12-00504-t001:** Baseline demographic, BMI percentile, and lipid parameters of the total population, and boys and girls when taken separately. Categorical variables are expressed as percentages and non-normal continuous variables are expressed as Median with its interquartile ranges (1st quartile–3rd quartile).

	Total Population (*n* = 681)	Boys (*n* = 347)	Girls (*n* = 334)
Age	12.94 (10.68–14.73)	13.21 (10.80–14.73)	12.54 (10.52–14.73)
Age groups			
8–11 years (*n* = 279)	40.97%	37.18%	44.91%
12–14 years (*n* = 243)	35.68%	39.19%	32.04%
15–18 years (*n* = 159)	23.35%	23.63%	23.05%
BMI percentiles			
Obese (*n* = 77)	11.31%	13.26%	9.28%
Overweight (*n* = 130)	19.09%	21.90%	16.17%
Normal or thinness (*n* = 474)	69.60%	64.84%	74.55%
Lipid parameters			
TC (mmol/L)	4.10 (3.60–4.60)	4.00 (3.54–4.50)	4.19 (3.60–4.60)
LDL-C (mmol/L)	2.50 (2.08–2.96)	2.45 (2.05–2.94)	2.56 (2.14–3.01)
TG (mmol/L)	1.08 (0.80–1.56)	1.05 (0.75–1.48)	1.14 (0.84–1.61)
HDL-C (mmol/L)	1.30 (1.10–1.50)	1.25 (1.10–1.50)	1.30 (1.10–1.50)
Non-HDL-C (mmol/L)	2.70 (2.30–3.20)	2.61(2.27–3.20)	2.80 (2.32–3.30)
Lp(a) (nmol/L)	26 (10–48)	25 (10–46)	26 (10–55)

**Table 2 metabolites-12-00504-t002:** The relationship between PCSK9 levels and age and Body Mass Index (BMI). The PCSK9 concentrations (ng/mL) are expressed as median with interquartile ranges (1st quartile–3rd quartile). The Mann–Whitney U test was used to calculate the *p* values of PCSK9 between boys and girls. The Kruskal–Wallis test yielded *p* values for age groups and BMI categories.

Characteristics	Total Population	Boys	Girls	*p*-Value (Gender)
Total population	63.22 (44.59–91.80)	65.14 (42.40–95.80)	62.05 (46.48–88.45)	0.60
Age groups				
8–11 years (*n* = 279)	64.30 (46.95–93.38)	71.08 (47.55–109.0)	59.63 (46.26–86.75)	0.07
12–14 years (*n* = 243)	63.22 (42.82–86.83)	63.62 (40.53–85.91)	63.16 (46.36–88.18)	0.91
15–18 years (*n* = 159)	60.55 (44.51–93.96)	58.70 (40.69–86.03)	61.74 (47.30–95.34)	0.35
*p*-value (age groups)	0.49	0.13	0.71	
BMI percentiles				
Obese	64.86 (46.60–90.45)	67.41 (48.34–99.83)	58.84 (44.76–85.71)	0.55
Overweight	68.30 (49.17–100.10)	73.80 (48.97–102.60)	62.67 (48.94–96.08)	0.39
Normal or thinness	61.56 (43.26–89.69)	60.02 (40.61–94.41)	62.25 (46.36–88.70)	0.75
*p*-value (BMI group)	0.13	0.15	0.76	

**Table 3 metabolites-12-00504-t003:** Baseline demographic, PCSK9 values, and lipid parameters of the five subgroups (high TC, high LDL-C, high Non-HDL-C, high Lp(a), high TG) when taken independently. Categorical variables are expressed as percentages and non-normal continuous variables are expressed as median with its interquartile ranges (1st quartile–3rd quartile).

	Normal/ Borderline Group (*n* = 628)	High TC (*n* = 53)	Normal/ Borderline Group (*n* = 594)	High LDL-C (*n* = 87)	Normal/ Borderline Group (*n* = 609)	High Non-HDL-C *n* = 72)	Normal/ Borderline Group (*n* = 593)	High Lp(a) (*n* = 88)	Normal/ Borderline Group (*n* = 483)	High TG (*n* = 198)
Age	12.95 (10.74–14.71)	12.72 (10.15–15.15)	12.95 (10.67–14.71)	12.94 (10.92–15.28)	12.94 (10.72–14.69)	13.00 (10.32–15.40)	12.90 (10.59–14.71)	13.31 (11.32–14.95)	13.18 (10.88–15.23)	12.33 (10.34–14.24)
PCSK9 value (ng/mL)	62.29 (44.05–89.17)	82.10 (55.72–120.6)	61.30 (43.98–88.25)	77.25 (53.60–119.10)	61.85 (43.99–89.31)	73.40 (57.18–118.60)	63.85 (45.31–92.13)	58.09 (41.15–84.92)	63.39 (45.07–90.96)	62.95 (43.85–96.05)
Lipid parameters										
TC (mmol/L)	4.00 (3.57–4.40)	5.60 (5.43–5.90)	4.00 (3.50–4.380)	5.30 (4.90–5.70)	4.00 (3.52–4.40)	5.48 (5.00–5.80)	4.10 (3.60–4.59)	4.10 (3.73–4.78)	4.00 (3.50–4.47)	4.30 (3.89–4.80)
LDL–C (mmol/L)	2.44 (2.06–2.84)	3.94 (3.59–4.20)	2.40 (2.04–2.79)	3.62 (3.45–4.05)	2.41 (2.05–2.81)	3.77 (3.55–4.10)	2.48 (2.07–2.95)	2.59 (2.19–3.10)	2.40 (2.04–2.86)	2.77 (2.29–3.25)
TG (mmol/L)	1.05 (0.78–1.50)	1.38 (1.09–1.91)	1.025 (0.77–1.47)	1.43 (1.10–1.92)	1.02 (0.77–1.44)	1.74 (1.22–2.45)	1.07 (0.79–1.57)	1.20 (0.84–1.54)	0.89 (0.71–1.13)	1.92 (1.63–2.46)
HDL-C (mmol/L)	1.30 (1.10–1.50)	1.30 (1.15–1.65)	1.30 (1.10–1.50)	1.20 (1.00–1.40)	1.30 (1.10–1.50)	1.10 (1.00–1.32)	1.30 (1.10–1.50)	1.30 (1.1–1.50)	1.30 (1.20–1.50)	1.10 (1.00–1.30)
Non-HDL-C (mmol/L)	2.64 (2.30–3.10)	4.30 (3.90–4.55)	2.60 (2.26–3.00)	4.00 (3.72–4.38)	2.60 (2.30–3.10)	4.10 (3.90–4.40)	2.70 (2.30–3.20)	2.90 (2.40–3.30)	2.60 (2.20–3.00)	3.11 (2.70–3.70)
Lp(a) (nmol/L)	24 (10–46.75)	44 (20–104)	24 (10–46.25)	35 (15–65)	25 (10–48)	32.50 (14.25–62)	21 (8–40)	130 (94–175)	25 (10–48)	29 (10–52)

**Table 4 metabolites-12-00504-t004:** Two separate multiple linear regression analyses for boys and girls with PCSK9 as a dependent variable and age, BMI, non-HDL-C, TG, HDL-C, and Lp(a) as independent variables. Due to their non-normal data, lipid parameters, and BMI were entered into the model using their natural logarithmic transform. BMI: Body Mass Index.

Variable	Β	Std. Error	*p*-Value
A-Boys			
Intercept	4.07	0.28	<0.0001
Age	−0.007	01	0.62
Ln (BMI)	−0.03	0.04	0.43
Ln (Non-HDL-C)	0.48	0.13	0.0003
Ln (TG)	0.02	0.08	0.84
Ln (HDL-C)	0.25	0.17	0.16
Ln (Lp(a))	−0.06	0.03	0.07
B-Girls			
Intercept	3.55	0.23	<0.0001
Age	0.02	0.01	0.11
Ln (BMI)	−0.02	0.03	0.52
Ln (Non-HDL-C)	0.48	0.12	0.0001
Ln (TG)	−0.01	0.07	0.86
Ln (HDL-C)	0.04	0.14	0.78
Ln (Lp(a))	−0.006	0.03	0.82

## Data Availability

All the data have been included in article.

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
