# Peer review of "Circulating PCSK9 Linked to Dyslipidemia in Lebanese Schoolchildren"

_metabolites, 2022, doi:10.3390/metabo12060504_

Round 1

Reviewer 1 Report

the manuscript is well written and numbers in both sexes is good.

Author Response

Minor spell check changes were performed

Reviewer 2 Report

The authors explored serum PCSK9 levels and the relationship between PCSK9 and the lipid profile in a sample of Lebanese school children. 

The manuscript is overall well-written and the study seems scientifically sound.

Several suggestions for revision are listed below:

  1. There is no need for capitalization in the title of the paper and in the subsections.
  2. Some sections of the Metabolites template have been deleted by mistake from the first page (e.g. date received, date submitted, etc.). In addition, the lines have been deleted which makes the evaluation of the paper more difficult.
  3. The paper requires a minor polishing of the English language. Please use the free version of Grammarly at least.
  4. "Non-consenting children" - shouldn't informed consent be sought from the caretakers/parents in the case of children? Were the subjects aged 18 or older? Are the regulations different in Lebanon? Please clarify. In the last section of the Methods, you state that consent was sought from their parents. Please clarify this ambiguity. 
  5. Please add the approval date/year for the ethics committee decision.
  6. "Non-HDL-C was chosen as an in-dependent variable because the sampling of children was performed in a non-fasting state and non-HDL-C has been recommended as a better meas-urement especially among subjects with high TG levels. In addition, non-HDL-C is a better estimation of the atherogenic lipoproteins." - please reference these statements.
  7. Decrease the fonts in the tables to improve readability.
  8. The presentation of P-values is incorrect. Please correct the presentation of the P-values as suggested in the following guideline paper: "Too many digits: the presentation of numerical data". P-values should be reported to two decimal places unless the first two are 00 or the number lies between 0.045 and 0.050. Please review all P-values: https://www.ncbi.nlm.nih.gov/pmc/articles/PMC4483789  
  9.  The authors put a lot of emphasis on familial hypercholesterolemia, but were any of these children suffering from this disorder?
  10. The discussions can be improved. The authors should also try and explain the correlations between PCSK9 and the components of the lipid profile. What do these associations mean?
  11. What is the clinical applicability of your study in pediatrics and cardiology? Should these children be screened for cardiovascular disease early? It would be interesting to see the relationship between metabolically healthy obese vs non-obese children in terms of PCSK9 levels. In addition, dyslipidemia favors the risk of CVD also via inflammation and oxidative stress. See the following papers previously published in Metabolites: https://www.mdpi.com/2218-1989/10/5/195
  12. The references are not written in the style required by Metabolites, namely the American Chemical Society. Moreover, the references should be cited in squared brackets and not round brackets, i.e., [1] not (1).

Author Response

  1. There is no need for capitalization in the title of the paper and in the subsections.

The capitalization was changed.

  1. Some sections of the Metabolites template have been deleted by mistake from the first page (e.g. date received, date submitted, etc.). In addition, the lines have been deleted which makes the evaluation of the paper more difficult.

These sections were added.  

  1. The paper requires a minor polishing of the English language. Please use the free version of Grammarly at least.

This was done by Grammarly.

  1. "Non-consenting children" - shouldn't informed consent be sought from the caretakers/parents in the case of children? Were the subjects aged 18 or older? Are the regulations different in Lebanon? Please clarify. In the last section of the Methods, you state that consent was sought from their parents. Please clarify this ambiguity.

Children aged 8 to 17 need a consent from their parents to participate in the study, subjects aged 18 signed the informed consent. This was clarified in the method section

  1. Please add the approval date/year for the ethics committee decision.

The approval dates are year 2013 for the publication reference 23 and year 2018 for the publication 25 and the current study. This was added to the manuscript.

  1. Non-HDL-C was chosen as an independent variable because the sampling of children was performed in a non-fasting state and non-HDL-C has been recommended as a better measurement especially among subjects with high TG levels. In addition, non-HDL-C is a better estimation of the atherogenic lipoproteins." - please reference these statements.

A reference (reference 32) was added for this sentence which is the following: “Frontini MG, Srinivasan SR, Xu J, Tang R, Bond MG, Berenson GS. Usefulness of childhood non-high density lipoprotein cholesterol levels versus other lipoprotein measures in predicting adult subclinical atherosclerosis: The Bogalusa Heart Study. Pediatrics. 2008;121: 924–929”.

  1. Decrease the fonts in the tables to improve readability.

This was done for Table 3.

  1. The presentation of P-values is incorrect. Please correct the presentation of the P-values as suggested in the following guideline paper: "Too many digits: the presentation of numerical data". P-values should be reported to two decimal places unless the first two are 00 or the number lies between 0.045 and 0.050. Please review all P-values: https://www.ncbi.nlm.nih.gov/pmc/articles/PMC4483789  

This was done.

  1. The authors put a lot of emphasis on familial hypercholesterolemia, but were any of these children suffering from this disorder?

The following sentence was added to the results’ section 3.2 with the reference 33:

19 children (2.8%) had very high LDL-C (≥ 4.1 mmol/L) suggestive of FH [33].

However, in these children, we did not collect information on their family history of premature coronary artery disease, or perform a DNA analysis to confirm the diagnosis of FH.

  1. The discussions can be improved. The authors should also try and explain the correlations between PCSK9 and the components of the lipid profile. What do these associations mean?

Several improvements were done to the discussion according to the reviewer’s remark. For example, the following sentence was added to the discussion's first paragraph:

PCSK9 is one of the ligands of LDLr; it uses it as an exit route from the plasma compartment while also inducing its degradation; thus, plasma PCSK9 levels, LDLr expression, and plasma LDL-C levels are all reciprocally regulated [34]. A new reference number 34 was added

This sentence was followed by the sentences previously discussed:

The PCSK9 half-life was shown to be much longer in LDLr -/- mice, while overexpression of hepatic LDLr decreases serum levels of murine PCSK9 [35]. The LDL particle also increases PCSK9 affinity for the LDLr by protecting its cleavage from furin; this results in a substantial decrease in plasma PCSK9 activity [34].

  1. What is the clinical applicability of your study in pediatrics and cardiology? Should these children be screened for cardiovascular disease early? It would be interesting to see the relationship between metabolically healthy obese vs non-obese children in terms of PCSK9 levels. In addition, dyslipidemia favors the risk of CVD also via inflammation and oxidative stress. See the following papers previously published in Metabolites: https://www.mdpi.com/2218-1989/10/5/195

  • A reference was added at the end of the discussion related to the link between PCSK9, inflammation, and atherosclerosis.
  • PCSK9 has been associated with inflammation in several studies as reviewed by Momtazi-Borojeni et al. Furthermore, the subendothelial accumulation of cholesterol caused by hypercholesterolemia leads to vascular inflammation [49].
  • Correlation between different BMI groups and PCSK9 was done. We did not find a significant correlation (Table2).

  1. The references are not written in the style required by Metabolites, namely the American Chemical Society. Moreover, the references should be cited in squared brackets and not round brackets, i.e., [1] not (1).

These modifications have been done.

Reviewer 3 Report

The authors present an interesting manuscript  in which they have studied potential correlations between the protein PCSK9 and lipids markerts of CVD in children. The manuscript is well-written, the conclussions are supported by the results and authors have made a very-good discussion.

Nevertheless, I have some comments for the authors.

  1. I think my biggest concern is the fact that it is not easy to know what results out the included in the manuscript have been already published and previously discussed by the authors. Thus, I think the authors must clearly state which data are already published, which one not. For instance, are all the data showed in table 1 already published and discussed? If so, it should be clearly declare. The same for the comments about the BMI distribution (end of section 3.1) and the results showed in section 3.2 and in table 2 (section 3.3). Moreover, in section 3.5. those data from table 1 are reported again. If they are alredy published, the reference should be included. The data from the menarche and testosterone values (section 3.6) are not clear enough if they are alredy published. Additionally, those data (number of girls with/whithout menarche and number of boys with detectable testosterone levels) are not included in the manuscript.

  1. In Discussion (page 11, second paragraph): Did the authors think that the PCSK9 determination could have any advantage over TC, LDL-c and non-HLD-C levels? Is it a more sensible/reliable/easy/cheaper/robust determination? May be used for the development of a sort of inmuno-tests for self detection? Does PCSK9 level provide any additional information?. I think the authors should highlight a bit more the importance for studying PCSK9 levels, especially if the rest of data show in the manuscript have been already published.

Minor comments:

In Material and Methods: Could the  authors explain in the present manuscript the criteria which the authors follow for the clasiffication of the children into three age groups?

“Samples were collected in 2013 and 2014 from non-fasting children be-tween 8 and 10 AM.”-> Blood samples.

“Non-consenting children, as well as children suffering from any acute or chronic medical condition (such as diabetes or hypothyroidism) or taking any drug that can affect the lipid profile (contraceptive pills, isotretinoin, oral corticosteroids, atypical antipsychotics, or immunosuppressive therapies), were all excluded from the study.” -> How many of them were excluded for these reasons?

In section 3.4., “… (r = 0,037, p=0.33; r = 0,038, p =0.49; and r = 0,047, p =0.39, respectively); Similarly, no significant correlation …-> Change ; by .

In Discussion: We didn’t find…; We also didn’t find …-> We did not.

All authors have read and agreed to the published version of the manuscri  -> cut sentence.

  1. In conclussion -> Conclussions

Author Response

Response to reviewer 3:

  1. I think my biggest concern is the fact that it is not easy to know what results out the included in the manuscript have been already published and previously discussed by the authors. Thus, I think the authors must clearly state which data are already published, which one not. For instance, are all the data showed in table 1 already published and discussed? If so, it should be clearly declare. The same for the comments about the BMI distribution (end of section 3.1) and the results showed in section 3.2 and in table 2 (section 3.3). Moreover, in section 3.5. those data from table 1 are reported again. If they are already published, the reference should be included. The data from the menarche and testosterone values (section 3.6) are not clear enough if they are already published. Additionally, those data (number of girls with/without menarche and number of boys with detectable testosterone levels) are not included in the manuscript.

 Results of BMI percentiles and lipid parameters were previously published in the overall sample of 969 school children published in 2016. Results presented here in table 1 and section 3.1, 3.2, and part of 3.5 are those of the subgroup as mentioned in the method section: “The population included in this study is a subgroup of our initial cohort”. Obviously, because the sample is smaller, the values are different. Because the purpose of the study was only to discuss the PCSK9 results, those results were not discussed. This explanation was added to the manuscript in the flow chart that was requested by the reviewer 4.

Results of section 3.3 and section 3.6 as well as table 2 were not previously published since they are related to PCSK9.

The number of girls with menarche was already added in the results’ section (3.1) and has been re-added in the results’ section (3.6).

The number of boys with detectable testosterone levels was added in the results’ section (3.6).

  1. In Discussion (page 11, second paragraph): Did the authors think that the PCSK9 determination could have any advantage over TC, LDL-c and non-HLD-C levels? Is it a more sensible/reliable/easy/cheaper/robust determination? May be used for the development of a sort of immuno-tests for self-detection? Does PCSK9 level provide any additional information? I think the authors should highlight a bit more the importance for studying PCSK9 levels, especially if the rest of data show in the manuscript have been already published.

The discussion was developed, and these two paragraphs were added.

  • It is noteworthy that another study has demonstrated that PCSK9 levels could be related to cardiovascular (CV) events independently of the levels of atherogenic lipoproteins [45]. Finally, Leander et al. showed that even after adjusting for known CV risk factors, the association between serum PCSK9 and CV risk remained, suggesting that PCSK9 levels may play a role in CV events apart from regulating LDL cholesterol [50]. 
  • Further prospective studies are needed on a larger scale in different populations to confirm if PCSK9 levels provide additional information in predicting CVD beyond the recognized CV factors.

Minor comments:

In Material and Methods: Could the authors explain in the present manuscript the criteria which the authors follow for the classification of the children into three age groups?

We rely on our previous study (ref 23) where the population was categorized into the same age groups. The use of the same age categorization allows us to look at the differences according to age group between the lipid profile (first publication), Lp(a) (second publication) (ref 25) and PCSK9.

“Samples were collected in 2013 and 2014 from non-fasting children be-tween 8 and 10 AM.”-> Blood samples.

The correction was done.

“Non-consenting children, as well as children suffering from any acute or chronic medical condition (such as diabetes or hypothyroidism) or taking any drug that can affect the lipid profile (contraceptive pills, isotretinoin, oral corticosteroids, atypical antipsychotics, or immunosuppressive therapies), were all excluded from the study.” -> How many of them were excluded for these reasons?

Those were the exclusion criteria. The sentence was rewritten accordingly. Very few children were excluded for these findings. The exact number was not recorded.

In section 3.4., “… (r = 0,037, p=0.33; r = 0,038, p =0.49; and r = 0,047, p =0.39, respectively); Similarly, no significant correlation …-> Change ; by .

This was done

In Discussion: We didn’t find…; We also didn’t find …-> We did not.

This was corrected

All authors have read and agreed to the published version of the manuscri  -> cut sentence.

This was corrected

  1. In conclussion -> Conclussions This was corrected

Reviewer 4 Report

To:

Editorial Board

Metabolites

Title: “CIRCULATING PCSK9 LINKED TO DYSLIPIDEMIA IN LEBANESE SCHOOLCHILDREN”

Dear Editor,

I read this paper and I think that:

  • It would be interesting to consider the impact of diet on PCSK9 levels. Scicchitano P et al. (Journal of Functional Foods 2014,6:11-32) demonstrated the impact of nutraceuticals on PCSK9 levels. Therefore, nutraceuticals could impact on PCSK9. Please discuss such a point.
  • Please revise the English of the paper in order to amend typos.
  • A flow chart of the study should be included.
  • The reproducibility of Lp(A) measurements should be provided.

Author Response

 Response to reviewer 4

It would be interesting to consider the impact of diet on PCSK9 levels. Scicchitano P et al. (Journal of Functional Foods 2014,6:11-32) demonstrated the impact of nutraceuticals on PCSK9 levels. Therefore, nutraceuticals could impact on PCSK9. Please discuss such a point.

Since the type of diet was not studied in our population, it is difficult to discuss this point. A sentence related to this point was added in the limitation section of the discussion with the Scicchitano P et al. reference.

 Please revise the English of the paper in order to amend typos.

This was done

A flow chart of the study should be included.

A flow chart was introduced.

It was done and added after the conclusion as a supplementary material.

The reproducibility of Lp(A) measurements should be provided

The intra and inter-coefficient of variation is less than 4%.

Round 2

Reviewer 2 Report

The authors have answered my comments in a satisfactory fashion and the paper can be accepted  for publication in Metabolites.

Reviewer 3 Report

I want to thank the authors for the effort made in addressing all my comments. I am satisfied with the changes.

Reviewer 4 Report

Dear Editor,

the  authors well addressed my previous comments. The paper improved very much.